# Meta-learning how to Share Credit among Macro-Actions

**Ionel-Alexandru Hosu**
Politehnica University of Bucharest
Bucharest, Romania
ionel.hosu@cs.pub.ro

**Traian Rebedea**
NVIDIA &
Politehnica University of Bucharest
Bucharest, Romania
trebedea@nvidia.com

**Razvan Pascanu**
Mila – Quebec Artificial Intelligence Institute
Montreal, Canada
r.pascanu@gmail.com

## Abstract

One proposed mechanism to improve exploration in reinforcement learning is through the use of macro-actions. Paradoxically though, in many scenarios the naive addition of macro-actions does not lead to better exploration, but rather the opposite. It has been argued that this was caused by adding non-useful macros and multiple works have focused on mechanisms to discover effectively environment-specific *useful* macros. In this work, we take a slightly different perspective. We argue that the difficulty stems from the trade-offs between reducing the average number of decisions per episode versus increasing the size of the action space. Namely, one typically treats each potential macro-action as independent and atomic, hence strictly increasing the search space and making typical exploration strategies inefficient. To address this problem we propose a novel regularization term that exploits the relationship between actions and macro-actions to improve the credit assignment mechanism by reducing the effective dimension of the action space and, therefore, improving exploration. The term relies on a similarity matrix that is meta-learned jointly with learning the desired policy. We empirically validate our strategy looking at macro-actions in Atari games, and the StreetFighter II environment. Our results show significant improvements over the RAINBOW-DQN baseline in all environments. Additionally, we show that the macro-action similarity is transferable to related environments. We believe this work is a small but important step towards understanding how the similarity-imposed geometry on the action space can be exploited to improve credit assignment and exploration, therefore making learning more effective.

## 1 Introduction

While Reinforcement Learning (RL) lead to a plethora of successes [e.g. 1–5], being an integral component of modern LLMs as well [e.g. 6], efficient exploration remains a significant challenge. This is particularly so in environments with large or complex action spaces or sparse rewards that require to do credit assignments over long episodes. One potential approach to simplify exploration is the use of macro-actions [e.g. 7–10], fixed sequences of actions that are frequently used by the policy. By expanding the action space with macro-actions, and using them efficiently, an agent can reduce considerably the number of decisions that it needs to take during an episode, hence making the exploration easier.

39th Conference on Neural Information Processing Systems (NeurIPS 2025).

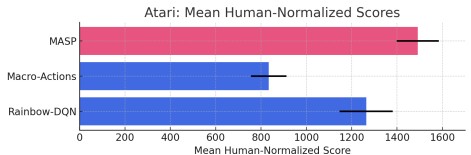
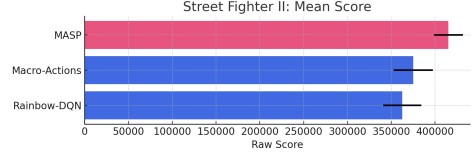

|  (a) Atari Average Human Normalized Scores  |  (b) Street Fighter II Scores  |

Figure 1: Visual comparison between *Rainbow-DQN*, *Rainbow-DQN + Macro-Actions* and *Macro-Action Similarity Penalty*

.

However it is often that even when one has access to a good proposal distribution for macro-actions, using them leads to worse performance. This is due to the fact that even if the average length of an episode might decrease, in many cases the exploration space actually becomes larger, as the number of available choices at each step can increase considerably (see also Figure 1a).

In this work, we attempt to exploit the fact that some of these macro-actions are inherently related, not only among themselves but also to the basic actions. By simply increasing the action space with additional macro-actions we are implicitly assuming that each macro-action is atomic and independent. But often, due to the sharing of basic actions between these sequences, the outcome of different macro-actions in the environment can be somewhat similar. We propose to capitalize on this similarity to distribute the credit received by a chosen macro-action to all other macro-actions according to their similarity. This effectively reduces the intrinsic dimensionality of the action space, therefore reducing the size of the search space and improving exploration.

The similarity between macro-actions acts as an inductive bias, which can be either handcrafted given a good understanding of the available macro-actions, or, as we will show in Section 3, it can be *meta-learned* from the data. For simplicity, in this work we assume that the similarity takes the form of a fixed kernel, $\Sigma$, that is used to push $Q$ values towards each other proportional to the similarity strength. Note that we make no explicit distinction between actions and macro-actions. We refer to this regularization term as the Macro-Action Similarity Penalty (MASP).

We apply this method in two scenarios. First, in Atari games, where macro-actions are extracted using a simple frequency-based heuristic over recorded trajectories, capturing common sequences of basic actions. The second scenario is the StreetFighter II environment, where each player has specific sets of action combos that result in more powerful attacks. We use these combos as potential macro-actions, incorporating all available combos regardless of the character used by the agent. Our experimental results demonstrate that the proposed method significantly improves learning efficiency on top of RAINBOW DQN [11], see Figure 1. Agents trained with our regularization approach exhibit faster convergence and higher cumulative rewards. By bridging the gap between extended action spaces and efficient exploration, this work contributes a novel approach that leverages the inherent geometry of the action space, potentially benefiting a wide range of reinforcement learning applications with structured action spaces.

## 2 Related work

Macro-actions, or temporally extended actions, have significantly contributed to improved exploration efficiency within reinforcement learning (RL) frameworks. Initial studies by Randlov [12] introduced systematic methods to construct macro-actions from primitive actions, accelerating learning in tasks such as bicycle balancing and grid-world navigation. Similarly, McGovern and Sutton [13] empirically confirmed their benefits, showing faster discovery of optimal solutions compared to primitive-level exploration.

**Macro-actions in Deep and Meta-RL.** Recent work has extended macro-actions to deep reinforcement learning settings. For example, Frans et al. [14] explored meta-learning strategies that enable hierarchical agents to generalize by reusing temporally extended skills. More recently, Cho and Sun [15] proposed a tri-level meta-RL framework that learns task-agnostic macro-actions across environments. Our work draws inspiration from such architectures but focuses on meta-learning

similarity structures among macro-actions to enhance credit assignment rather than hierarchical decomposition alone.

**Macro-actions in Multi-agent and Decentralized RL.** In decentralized and multi-agent settings, macro-actions have proven effective for improving exploration and coordination. Tan et al. [16] proposed the Macro Action Decentralized Exploration Network (MADE-Net) to deal with communication dropouts in cooperative multi-agent environments. Similarly, Xiao [17] introduced both decentralized and centralized macro-action value learning approaches, showing scalability and coordination gains.

**Intrinsic Motivation and Macro-actions.** Macro-actions have also been combined with intrinsic motivation to promote deeper exploration. Panda et al. [18] presented a framework blending intrinsic and extrinsic rewards across timescales, enabling agents to develop strategic behaviors involving macro-action.

**Transferability and Reusability.** Macro-actions are particularly valuable for transfer learning. Chang et al. [19] demonstrated that macro-actions are reusable across tasks and algorithms and transferable to new environments with different reward functions. Such findings suggest that macro-actions can encode abstract behaviors robust to minor environment changes.

**Planning with Macro-actions.** Outside RL, macro-actions have been leveraged in classical planning. Allen et al. [20] proposed focused-effect macro-actions to improve heuristic accuracy in black-box planning, while Newton et al. [21] integrated macro-actions into planning heuristics, showing better performance in relaxed planning graph search. These results emphasize the role of macro-actions not only in learning but also in accelerating symbolic planning through structured action abstractions.

**Macro-actions for POMDPs.** To deal with partial observability, Amato et al. [22] studied macro-actions in decentralized POMDPs, proposing solutions that integrate temporally abstracted behaviors into tree-based planners. Lee et al. [23] introduced MAGIC, which uses learned macro-actions to improve online planning in POMDPs by tailoring the action space to situation-aware behavior.

**Hierarchical and Object-Oriented Macro-Actions.** In the context of spatial reasoning, Hakenes and Glasmachers [24] proposed object-oriented macro-actions based on topological maps, which provide inductive structure for learning in visually complex and sparse environments. These topological abstractions act as priors for exploration and planning.

**Our Contribution.** Despite these advances, integrating macro-actions in a scalable and flexible way remains challenging, particularly when they are treated as independent and the relationships among them are ignored. Our approach fills this gap by introducing a meta-learned similarity matrix for macro-actions, encouraging credit sharing across related actions through a regularization mechanism. This not only improves exploration in large action spaces but also adapts the agent's inductive biases during training, aligning with the broader goals of structure-aware learning in RL.

## 3  Methods

### 3.1  Background

We assume the standard MDP formulation, where $\mathcal{M} = (\mathcal{S}, \mathcal{A}, P, r, \rho_{init}, \gamma)$ consisting of state space $\mathcal{S}$, the set of all possible actions $\mathcal{A}$, a transition distribution $P(s'|s, a)$ and a reward function $r : \mathcal{S} \times \mathcal{A} \rightarrow \mathbb{R}$. The goal of an agent is to find a policy $\pi$ that maximises the expected sum of discounted rewards: $Q^{\pi}(s, a) = \mathbb{E}_{s_0 \sim \rho_{init}, a_t \sim \pi(s_t)} [\sum_{t=0} \gamma^t r(s_t, a_t)]$. In our work we focus on value based methods, which minimize the *temporal difference* (TD) error in order to learn the $Q$-function and define $\pi$ greedily with respect to it. We are specifically assuming that $Q$ is approximated by a neural network. The prototypical algorithm is DQN [25], where $Q$ is learned by minimising:

$$\mathcal{L}(\theta) = \mathbb{E}_{(s,a,r,s') \sim \mathcal{D}} \Big( Q_\theta(s, a) - \big(r + \gamma \max_{a'} Q_{\theta'}(s', a')\big) \Big)^2 \tag{1}$$

Multiple works, after the seminal DQN paper [25] had improved this algorithms, as for example agent C51 [26] who instead of modeling the return directly, predicts a categorical distribution of expected returns. RAINBOW [27] includes several improvements on top of C51, ranging from prioritised experience replay [28] to $n$-step returns for lower variance bootstrap targets, double Q-learning [29], disambiguation of state-action value estimates [30], and noisy nets [31]. In our work we use

RAINBOW as our baseline and build on top of it by augmenting the action space with Macro-Actions and explore our proposed regularization term MASP.

## 3.2 MASP: Macro-Actions Similarity Penalty

One specific intuition behind extended actions, shared with concepts like macro-actions or options, is that they reduce the number of decisions that we need to take within the episode, and hence make learning easier (from a credit assignment point of view). Simply put, having fewer decisions in an episode means that it is easier to see which of them contributed to the observed return.

Another important element of learning in RL, however, is that of exploration. And while shorter episodes should make exploration easier, when we consider an extended action space, the outcome can be that exploration becomes considerably harder, given that the number of choices per step increases. One intuitive way of understanding this behavior is to consider the size of the search space that the RL algorithm has to explore. If we make the assumption that for any given state, any of the actions leads to a *unique* outcome (i.e. different state), then the search space size should have the form $f^N$, where $f$ is the branching factor, or number of actions per step, and $N$ is the number of decisions the agent needs to take. One can now easily see that $N' < N$ does not guarantee that $f'^{N'} < f^N$ if reducing $N'$ requires increasing the branching factor, i.e. $f' > f$ which the usual situation when adding macro-actions to the action space. Indeed, it is more likely that increasing the branching factor will increase the search space faster then decreasing the number of decisions taken per episode.

The main assumption in the above reasoning is that each new action added to the action space is *unique* or leads to a *unique* outcome — assumption typically made by standard exploration strategies like $\epsilon$-greedy. However for typical environments this is not the case. The search space is highly structured, which is the main reason we can learn reasonable policies over extremely-large search spaces. This structure is typically exploited in deep RL through the use of neural networks as function approximators. In this work we exploit this further, specifically looking at the structure of the action space. We make the assumption that certain actions lead, on average, to similar outcomes. This will naturally impose a clustering of actions, and allow learning and exploration to first identify the correct cluster for a given state and then the fine-grained differences between actions within each cluster.

We propose to codify this intuition into a regularization term we refer to as the *Macro-Action Similarity Penalty* (MASP). The penalty enforces that any cluster of *similar* actions can not disperse too much in their Q-values, and, more importantly, learning has to move the entire cluster together, allowing Q-values to *learn* from each other. Crucially, the term allows for some dispersion which can capture some degree of differentiation between actions, which is needed to learn a reliable policy.

## 3.3 Detailed Formulation

Formally, we assume the existence of a similarity matrix, $\Sigma \in \mathbb{R}^{|\mathcal{A}| \times |\mathcal{A}|}$, where each element $\Sigma_{ij}$ represents the similarity between actions $a_i$ and $a_j$, marginalized over all possible states. Specifically, high $\Sigma_{ij}$ encodes the inductive bias that $Q_i$ and $Q_j$ should not differ substantially, regardless of state. The similarity matrix is symmetric ($\Sigma_{ij} = \Sigma_{ji}$) and non-negative.

To operationalize MASP, we incorporate it into the standard Temporal Difference (TD) loss as an additive term weighted by a hyper-parameter $\eta$ which decides how much we restrict the Q-values to respect $\Sigma$. Given a sampled batch of transitions $(s_i, a_i, r_i, s_{i+1})_{i=1}^n$, we have:

$$\mathcal{L}_{\text{MASP}} = \eta \cdot \|Q(s_i, \cdot; \theta) - \Sigma Q(s_i, \cdot; \theta)\|_2^2 \tag{2}$$

Intuitively, this loss penalizes large differences between the Q-values of similar actions, promoting a smoother and more structured Q-function landscape.

## 3.4 Meta-learning $\Sigma$

The efficacy of our approach depends on choosing an adequate similarity matrix $\Sigma$. For example, setting $\Sigma$ to the identity matrix, $\mathbb{I}$, will reduce MASP to not have any effect. This is true if we also set $\Sigma_{ij} = 1$ for all $i$ and $j$. While in principle one can use domain knowledge to define clusters of similar actions, in this work we rely on meta-learning approaches to learn directly from data a

relevant clustering during exploration. Note that $\Sigma$ is learned jointly with $\theta$, therefore being able to track the current policy $\pi_\theta$, rather than being forced to marginalize over policies.

Specifically, we apply a meta-gradient framework inspired by Xu et al. [32] to automatically infer the entries of $\Sigma$. Following the meta-gradient methodology, we alternate between two phases:

- In the first phase, we apply a standard RL update to the agent parameters $\theta$ using a sampled trajectory $\tau$, under the current similarity metric $\Sigma$. This yields updated parameters $\theta'$.

- In the second phase, we evaluate the quality of the new $\theta'$ on a held-out trajectory $\tau'$, using a differentiable meta-objective $J'(\tau', \theta')$, where $J'$ is just the TD-error term. Note that $J'$ is a function of $\Sigma$ only through $\theta'$ due to the update done on the augmented objective $J$ containing our MASP regularization term (see Algorithm 1 for details). (We clone the network parameters to obtain $\theta'$ after a simulated update step, as required for meta-gradient computation [32].)

The meta-gradient $\nabla_\Sigma J'$ is computed using the chain rule as follows:

$$\frac{\partial J'(\tau', \theta')}{\partial \Sigma} = \frac{\partial J'}{\partial \theta'} \cdot \frac{d\theta'}{d\Sigma},$$

and the trace-based approximation [32] allows for efficient online accumulation of $\frac{d\theta'}{d\Sigma}$. This gradient is then used to update $\Sigma$ using stochastic gradient descent, thereby enabling the agent to dynamically learn which macro-actions should be treated as functionally similar. The precise implementation, including hyperparameter settings and gradient handling, will be made publicly available with our code. Additional details about the proposed MASP regularization and its implementation can be found in Appendix A.

---

**Algorithm 1** Meta-learning Credit Assignment with MASP

---

**Require:** Replay buffer $\mathcal{D}$, macro-actions set $\mathcal{A}_{\text{macro}}$, initial network parameters $\theta$, target network parameters $\theta^-$, similarity matrix $\Sigma$ (meta-parameters), learning rates $\alpha$ and $\beta$, regularization coefficient $\eta$
1: Initialize Q-network $Q(s, a; \theta)$ and target network $Q^-(s, a; \theta^-)$
2: **for** each environment step $t = 1$ to $T$ **do**
3:     Select action $a_t$ using $\epsilon$-greedy over $Q(s_t, a; \theta)$
4:     Execute $a_t$, observe $r_t$, $s_{t+1}$
5:     Store transition $(s_t, a_t, r_t, s_{t+1})$ in buffer $\mathcal{D}$
6:     **if** time to update **then**
7:         Sample trajectory $\tau = \{(s_i, a_i, r_i, s_{i+1})\}_{i=1}^n$ from buffer $\mathcal{D}$
8:         Compute similarity embedding $e_\Sigma$ from $\Sigma$ via a learned projection
9:         Compute target: $y_i = r_i + \gamma \max_{a'} Q^-(s_{i+1}, a'; \theta^-)$
10:        Compute standard TD loss: $\mathcal{L}_{\text{TD}} = \frac{1}{n} \sum_i (Q(s_i, a_i; \theta) - y_i)^2$
11:        Compute MASP regularization: $\mathcal{L}_{\text{MASP}} = \eta \cdot \|Q(s_i, \cdot; \theta) - \Sigma Q(s_i, \cdot; \theta)\|_2^2$
12:        Update network parameters: $\theta \leftarrow \theta - \alpha \nabla_\theta (\mathcal{L}_{\text{TD}} + \mathcal{L}_{\text{MASP}})$
13:        // Meta-gradient phase
14:        Clone network parameters: $\theta' \leftarrow \theta$
15:        Sample second trajectory $\tau'$ from buffer $\mathcal{D}$
16:        Compute meta-objective $\mathcal{L}_{\text{meta}} = \mathcal{L}_{\text{TD}}(\theta')$ on $\tau'$
17:        Compute meta-gradient: $\nabla_\Sigma \mathcal{L}_{\text{meta}}$ via chain rule
18:        Update $\Sigma \leftarrow \Sigma - \beta \nabla_\Sigma \mathcal{L}_{\text{meta}}$
19:     **end if**
20:     **if** time to update target network **then**
21:         $\theta^- \leftarrow \theta$
22:     **end if**
23: **end for**

---

To cope with the non-stationarity introduced by the evolving similarity metric, we condition the value function and policy on $\Sigma$ using a low-dimensional embedding $e_\Sigma$, akin to Universal Value Function Approximators (UVFA) [33]. This enables the network to generalize across changes in the structure of $\Sigma$ and adapt smoothly during learning.

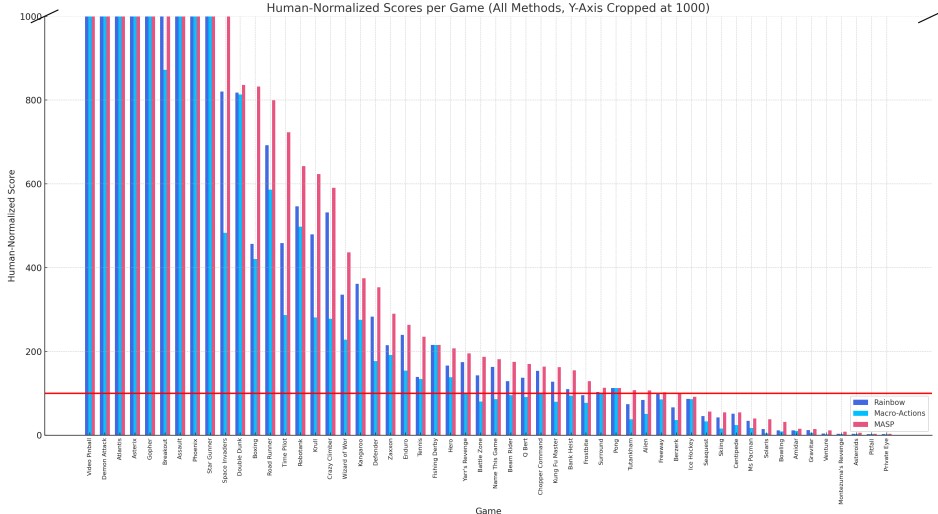

Figure 2: Visual comparison between *Rainbow-DQN* (dark blue), *Rainbow-DQN + Macro-Actions* (light blue), and *Macro-Action Similarity Penalty* (red).

**Learning the Similarity Embedding** $e_\Sigma$. A common and effective approach for representing the similarity matrix $\Sigma$ is to learn a low-dimensional embedding $e_\Sigma$ jointly with the main network parameters. Instead of using the full (potentially large) matrix $\Sigma$ directly as network input, we first flatten $\Sigma$ and map it to a compact vector via a trainable linear transformation:

$$e_\Sigma = W_{\mathrm{emb}} \cdot \mathrm{vec}(\Sigma),$$

where $W_{\mathrm{emb}}$ is a learned weight matrix and $\mathrm{vec}(\Sigma)$ denotes the flattened version of $\Sigma$. In our setup, $W_{\mathrm{emb}}$ is only updated via gradients from the main loss $J$, and not directly from the meta-objective $J'$. See Appendix A for more implementation details.

The resulting embedding $e_\Sigma$ is then concatenated with other input features (such as the state representation) and fed into the network as additional context. During training, $e_\Sigma$ is learned end-to-end via backpropagation, so the agent can flexibly adapt its notion of action similarity as $\Sigma$ evolves. This approach allows the network to capture the most relevant information about action similarities, supporting both efficient exploration and generalization, and is preferable to static or randomly projected embeddings when $\Sigma$ contains task-relevant structure.

This meta-learning mechanism improves credit assignment by promoting shared learning signals among semantically related actions – especially macro-actions, and leads to a more efficient exploration of the macro-action space. It also enables the agent to adapt its notion of action similarity as the environment dynamics and learning context evolve.

## 4 Experiments

| Game | Rainbow-DQN | +Macro-Actions | +MASP |
|---|---|---|---|
| **Breakout** | $379.5 \pm 25.1$ | $252.9 \pm 22.2$ | **884.4 ± 74.0** |
| **Frostbite** | $4,141.1 \pm 175.4$ | $3,361.2 \pm 356.4$ | **5,566.7 ± 317.3** |
| **Ms Pacman** | $2,570.2 \pm 204.6$ | $1,471.5 \pm 185.3$ | **2,966.5 ± 360.5** |
| **Seaquest** | $19,176.0 \pm 1,428.9$ | $13,917.1 \pm 641.1$ | **23,768.8 ± 1,476.9** |
| **Space Invaders** | $12,629.0 \pm 1,413.2$ | $7,496.4 \pm 532.9$ | **16,668.2 ± 2,050.5** |

Table 1: Comparison between RAINBOW DQN [11], RAINBOW-DQN with macro-actions (+Macro-Actions), and RAINBOW DQN with macro-actions similarity penalty (+MASP). **Bold** indicates maximal raw performance between RAINBOW DQN and MASP. Human-normalized (HN) scores are shown in parentheses. Cells highlighted in pink denote HN $\geq 100$. **This table is a cropped version for readability; the full results across all games are provided in the Appendix.**

| Game | # of macro-actions | +Macro-Actions | +MASP |
|---|---|---|---|
| **Breakout** | 64 | 248.1 ± 13.9 | **772.9 ± 54.9** |
| | 128 | 211.7 ± 9.6 | **727.0 ± 17.0** |
| | 256 | 147.6 ± 9.6 | **564.9 ± 26.1** |
| | 512 | 87.2 ± 3.9 | **482.9 ± 27.8** |
| | 1024 | 13.2 ± 0.5 | **379.5 ± 30.1** |
| **Frostbite** | 64 | 3,142.2 ± 142.7 | **5,204.2 ± 401.7** |
| | 128 | 2,853.5 ± 40.0 | **5,174.9 ± 246.7** |
| | 256 | 1,492.1 ± 100.3 | **4,446.6 ± 314.7** |
| | 512 | 893.7 ± 67.5 | **3,869.5 ± 297.9** |
| | 1024 | 470.1 ± 26.8 | **2,477.4 ± 188.7** |
| **Ms Pacman** | 64 | 1,752.1 ± 52.2 | **2,445.5 ± 35.9** |
| | 128 | 1,569.4 ± 58.3 | **2,144.3 ± 38.4** |
| | 256 | 1,129.6 ± 71.6 | **1,977.3 ± 140.9** |
| | 512 | 814.8 ± 56.4 | **1,278.5 ± 110.3** |
| | 1024 | 246.0 ± 18.8 | **933.4 ± 75.4** |
| **Seaquest** | 64 | 13,043.7 ± 576.0 | **21,966.7 ± 1065.0** |
| | 128 | 13,164.1 ± 599.8 | **18,766.7 ± 1550.6** |
| | 256 | 12,453.5 ± 734.7 | **16,897.4 ± 802.5** |
| | 512 | 8,332.6 ± 709.0 | **13,854.9 ± 315.4** |
| | 1024 | 4,333.8 ± 198.9 | **9,144.7 ± 723.1** |
| **Space Invaders** | 64 | 7,217.1 ± 451.9 | **15,566.9 ± 846.2** |
| | 128 | 8,325.0 ± 389.1 | **15,196.3 ± 1167.8** |
| | 256 | 6,913 ± 583.7 | **13,778.0 ± 683.1** |
| | 512 | 4,759.3 ± 170.8 | **9,769.3 ± 130.4** |
| | 1024 | 1,468.4 ± 49.5 | **6,793.5 ± 314.3** |

Table 2: Ablation study showing the effect of varying the number of macro-actions ($k \in 64, 128, 256, 512, 1024$) on performance across five representative Atari games. While the baseline with macro-actions suffers from severe performance degradation as the action space grows, MASP maintains strong and stable performance, demonstrating its robustness to action space size and ability to leverage structural similarity.

| Task | Rainbow DQN | +Macro-Actions | +MASP |
|---|---|---|---|
| **Macro-actions I** | 187,556 ± 3713.3 | 184,789 ± 7628.2 | **236,943 ± 13676.0** |
| **Macro-actions II** | 537,388.8 ± 23829.9 | 565,319.2 ± 9441.2 | **593,784 ± 18262.9** |

Table 3: Performance comparison in the StreetFighter II environment across two macro-action sets of increasing complexity. MASP consistently outperforms both the RAINBOW-DQN baseline and its macro-action-augmented variant. Notably, gains are amplified in the more complex Macro-Actions II setting, highlighting MASP's ability to scale with the size of the extended action spaces.

| Game | $P(\text{replace})$ | +Macro-Actions | +MASP |
|---|---|---|---|
| **Breakout** | 0 | 252.9 ± 22.2 | **884.4 ± 74.0** |
| | 0.25 | 208.3.3 ± 22.6 | **814.3 ± 37.7** |
| | 0.5 | 186.5 ± 12.9 | **663.9 ± 28.2** |
| | 0.75 | 76.4 ± 4.8 | **503.7 ± 39.6** |
| **Frostbite** | 0 | 3,361.2 ± 356.4 | **5566.7 ± 317.3** |
| | 0.25 | 3,071.9 ± 188.7 | **4,855.2 ± 353.7** |
| | 0.5 | 2,173.4 ± 83.7 | **4,194.0 ± 343.3** |
| | 0.75 | 1,366.3 ± 60.4 | **3,472.9 ± 272.9** |
| **Ms Pacman** | 0 | 1,471.5 ± 185.3 | **2966.5 ± 360.5** |
| | 0.25 | 1,145.9 ± 108.4 | **2,454.2 ± 255.9** |
| | 0.5 | 893.3 ± 87.9 | **2,144.2 ± 212.6** |
| | 0.75 | 666.5 ± 65.8 | **1,884.5 ± 153.8** |
| **Seaquest** | 0 | 13,917.1 ± 641.1 | **23768.8 ± 1476.9** |
| | 0.25 | 12,593.0 ± 250.4 | **21,619.8 ± 1586.2** |
| | 0.5 | 9,194.8 ± 562.5 | **19,913.5 ± 1296.9** |
| | 0.75 | 3,732.0 ± 172.1 | **17,581.0 ± 234.9** |
| **Space Invaders** | 0 | 7,496.4 ± 532.9 | **16,668.2 ± 2050.5** |
| | 0.25 | 6,692.6 ± 865.0 | **15,014.1 ± 1271.0** |
| | 0.5 | 3,836.2 ± 727.3 | **13,021.7 ± 238.1** |
| | 0.75 | 2,784.8 ± 116.0 | **10,481.9 ± 702.6** |

Table 4: Performance under macro-action noise. Each macro-action is replaced with a random sequence of the same length with probability $P(\text{replace}) \in 0.25, 0.5, 0.75$. MASP remains consistently effective across all noise levels, while RAINBOW-DQN + Macro-Actions degrades sharply, indicating MASP's robustness to imperfect or corrupted macro-action sets.

| Target Env. | Source of $\Sigma$ | | | | RAINBOW DQN |
| | Breakout | Montezuma's Revenge | Private Eye | Space Invaders | |
| --- | --- | --- | --- | --- | --- |
| **Breakout** | **884.4 ± 74.0** | 613.2 ± 57.1 | 884.4 ± 74.0 | 815.3 ± 66.2 | 379.5 ± 25.1 |
| **Montezuma's Revenge** | 162.1 ± 12.5 | **400.0 ± 18.1** | 376.8 ± 19.2 | 213.6 ± 11.4 | 154.0 ± 16.9 |
| **Private Eye** | 1,432 ± 117.4 | 2,074 ± 216.8 | **2244.3 ± 150.9** | 1,311.2 ± 138.5 | 1,704.4 ± 41.7 |
| **Space Invaders** | 15,294 ± 2,169.3 | 7,869.4 ± 614.5 | 7,483.7 ± 544.8 | **16668.2 ± 2050.5** | 12,629.0 ± 1,413.2 |

Table 5: Transfer learning results. Each column denotes the environment used to train (source) $\Sigma$, and each row is the evaluation (target) environment. Diagonal elements represent transfer of $\Sigma$ from the same game to itself; off-diagonals are transferred $\Sigma$ from different games. Last column is original RAINBOW performance for reference. Note that RAINBOW DQN with macro-actions and without MASP underperforms, hence why it was not added to the table.

**Experimental Setup.** We rigorously evaluate our proposed Macro-Action Similarity Penalty (MASP) across a diverse and challenging set of environments, including a suite of Atari 2600 games from the Arcade Learning Environment (ALE) [34] and the complex, structured action environment of StreetFighter II from Gym Retro [35]. These benchmarks represent scenarios with varied complexity and exploration demands, making them ideal for assessing the benefits of our approach.

All agents are trained for a substantial 2B frames in Atari and 500 million frames in StreetFighter II, using standard preprocessing steps and hyperparameters tuned per game via validation sweeps. Crucially, all agent variants share identical architectures and optimization hyperparameters, differing only in the application of macro-actions and the MASP regularization.

**Implementation Details.** For the Atari experiments, we build upon the RAINBOW-DQN [11] baseline, a robust and widely recognized reinforcement learning algorithm. Macro-actions were identified via a simple frequency-based heuristic, capturing frequently repeated sequences of primitive actions within successful trajectories. To thoroughly demonstrate MASP's effectiveness, we compare three agent variants: (1) standard RAINBOW-DQN, (2) RAINBOW-DQN augmented with independent macro-actions, and (3) RAINBOW-DQN enhanced with our proposed MASP method, which leverages the meta-learned similarity matrix. In Appendix B we provide all training hyperparameters and additional details for reproducibility.

**Atari Macro-Actions.** To construct meaningful macro-actions in Atari, we leverage the Atari Grand Challenge Dataset [36], which provides human expert trajectories. We extract the top-$k$ most frequent action subsequences of length 3 to 8, where $k = 32$ for the main results reported in Table 8. These action subsequences serve as candidate macro-actions and are appended to the base primitive action set to form the augmented action space.

**Results and Analysis.** Table 8 provides compelling evidence for MASP's advantage over both baselines. Notably, MASP consistently achieves higher cumulative rewards and demonstrates faster convergence across almost all Atari games tested. In many games, MASP dramatically improves upon RAINBOW-DQN, achieving performance that exceeds human-level benchmarks (a human-normalized score of at least 100). This improvement confirms our hypothesis: explicitly leveraging macro-action similarity significantly enhances credit assignment and exploration efficiency. For detailed experimental results see Appendix C.

**Ablation on Macro-Action Size.** We explore how MASP and baselines behave as the number of macro-actions increases. As shown in Table 2, we sweep $k \in 64, 128, 256, 512, 1024$ and observe that while RAINBOW + macro-actions suffer catastrophically as $k$ increases - presumably due to increased action space complexity - MASP maintains strong performance throughout. This highlights MASP's robustness and effectiveness in managing large action spaces.

**Robustness to macro-action reordering.** Another important clarification has to do with macro-action reordering. If the macro-actions are reordered, and the same permutation is applied to the rows and columns of $\Sigma$, the MASP regularization remains mathematically identical. This is because MASP only enforces that Q-values for actions with high similarity (as specified in $\Sigma$) are close — it does not rely on any fixed semantic meaning being attached to a particular index. In practice, we may choose to order macro-actions consistently (e.g., grouped by length or composition) solely to aid human interpretability and visualization, but this ordering is not required for the algorithm and

does not influence performance. The learned structure in $\Sigma$ generalizes across environments due to behavioral similarity between macro-actions, not index alignment.

**Noisy Macro-Actions.** To test MASP's robustness to imperfect macro-action sets, we conduct controlled noise ablation experiments. Specifically, for a fixed $k = 32$ macro-action set, we randomly replace each macro-action with a random sequence of the same length with a probability $P(\text{replace}) \in \{0.25, 0.5, 0.75\}$. This simulates conditions where the macro-action set is partially or entirely corrupted. As shown in Table 4, while Rainbow + Macro-Actions degrades sharply as noise increases, MASP remains remarkably stable and continues to provide substantial performance benefits, even at high noise levels. This suggests that the similarity structure learned by MASP is resilient and can cluster together non-informative macro-actions, learning easily to ignore them.

**Dependency on the quality of Macro-actions.** Another important point to note is that MASP relies on the initial macro-action pool being at least partially meaningful. Automatic discovery of high-quality macro-actions remains an open problem. That said, Table 4 shows that MASP remains robust even when 75% of macro-actions are randomly corrupted — suggesting the meta-learned similarity structure is resilient to noise and can learn to attenuate the influence of irrelevant macros. Future work could combine MASP with recent macro-action discovery methods (e.g., trajectory clustering, unsupervised option discovery), where MASP might serve as a regularizer to prune or refine an evolving macro pool.

**StreetFighter II Results.** For StreetFighter II, we utilize domain knowledge to define macro-actions explicitly as attack combinations (combos). Two distinct sets of macro-actions are considered: simpler (Macro-Actions I) and more complex combos (Macro-Actions II), testing MASP's adaptability and scalability across varied levels of action complexity. Table 3 highlights MASP's versatility. In the Macro-actions II scenario (complex combos), MASP significantly outperforms both the standard RAINBOW-DQN and Macro-Actions baselines, demonstrating clear benefits in structured, combinatorially large action environments. Even in simpler scenarios (Macro-actions I), MASP at least matches the best baseline, affirming its adaptability and broad utility.

**Transfer Learning and Generalization.** An additional exciting finding (see Table 5 and Appendix C for more detailed results) indicates that the learned macro-action similarity matrix $\Sigma$ generalizes well to related tasks, suggesting that MASP not only improves individual task performance but also facilitates knowledge transfer across related environments. Here we keep $\Sigma$ frozen to the one learned on the initial task (source domain), and relearn the policy with macro-actions under the frozen similarity metric. This transferability has profound implications for practical applications, where retraining from scratch for each new task is costly or infeasible. Notably, the results show strong transferability of the similarity matrix $\Sigma$ between games with similar structure and mechanics, such as between Montezuma's Revenge and Private Eye or between Breakout and Space Invaders, where performance remains close to or above the in-domain baseline. In contrast, transferring $\Sigma$ between more distinct games leads to a pronounced drop in performance, indicating that the benefits of transfer are greatest when the source and target environments share underlying dynamics or action semantics.

In Table 6, we show that $\Sigma$ generalizes best across environments with similar transition dynamics and interaction semantics (e.g., Breakout $\rightarrow$ Space Invaders, Montezuma $\rightarrow$ Private Eye). This supports the view that $\Sigma$ captures behavioral regularities across macros, rather than task-specific overfitting.

Our intuition is that although $\Sigma$ is not conditioned on state, it meta-learns how Q-values should co-vary across actions that yield similar *trajectory-level effects*. For example, in Breakout or Space Invaders, many macros correspond to repeated fire-move patterns. Thus, $\Sigma$ learns to group such macros — and this grouping remains useful when transferred to related games where the macro-level structure of agent-environment interactions is similar. This is analogous to how convolutional filters in vision models can transfer even though they are not image- or task-specific: they encode generic structure that is often reused.

**Interpretability.** Regarding interpretability, we visualize $\Sigma$ (Figure 3 in the Appendix) and observe clear block structures corresponding to groups of macros with similar behavioral outcomes. Still, we agree that $\Sigma$ is not trivially interpretable in semantic terms, and improving this is an exciting direction — e.g., via structured priors, sparse constraints, or clustering-based analyses.

**Discussion.** Collectively, these results robustly demonstrate MASP's value as a general-purpose technique that substantially improves learning efficiency, exploration, and final performance. Our extensive experimental validation across diverse environments emphasizes MASP's potential to significantly advance reinforcement learning methods that utilize macro-actions, highlighting the importance and effectiveness of intelligently structured credit assignment.

## 5   Limitations.

While our approach demonstrates significant empirical improvements in exploration and credit assignment through the use of macro-action similarity regularization, several limitations remain. We hope future research will address these limitations by exploring scalable approximations, improved macro-action discovery, and broader algorithmic applicability.

**Parametrization of $\Sigma$.** Our choice of parametrization makes $\Sigma$ independent of state, and of current policy $\pi_\theta$ (though there is a dependency on $\pi$ through the meta-learning process, as $\Sigma$ evolves jointly and possibly tracks $\pi$). Conditioning $\Sigma$ on state or $\theta$ could capture the structure of the action space better, though might make the meta-learning task considerably harder.

**Scalability to Very Large Action Spaces.** Although MASP improves robustness to the number of macro-actions, its computational cost grows quadratically with the size of the action space, as the similarity matrix $\Sigma$ must be maintained and projected at each update. Scaling to extremely large or continuous action spaces may require approximate or structured representations of similarity. Storing and projecting a full $\Sigma \in \mathbb{R}^{|\mathcal{A}| \times |\mathcal{A}|}$ can become expensive for extremely large action spaces. To mitigate this, we already use a low-dimensional learned embedding of $\Sigma$ (see Appendix A.3), which is passed to the Q-network as a context vector. While the full matrix is still used during regularization, it is only applied over the batch of Q-values at each update (not the entire action space per state). This allows for practical efficiency even with 1024+ actions, as shown in Table 2. In future work, we plan to investigate structured or sparse approximations (e.g., low-rank, kernel-based) to make MASP scalable to continuous or combinatorial large action spaces.

**Quality of Macro-Action Extraction.** The success of MASP relies on having a set of meaningful macro-actions. While we use a frequency-based heuristic on human trajectories, the performance may degrade if the extracted macro-actions are not relevant or are poorly aligned with the task structure. Automatic or adaptive discovery of optimal macro-actions remains an open problem.

**Interpretability and Generalization of Similarity Structure.** Although we observe some transferability of the learned similarity matrix across tasks, its interpretability and generalization properties are not fully understood. Further work is needed to analyze when and why meta-learned similarities capture useful domain knowledge.

**Computational Overhead.** Incorporating meta-learning and the additional regularization term introduces extra computational cost per training iteration compared to standard DQN baselines. In resource-constrained settings, this may limit applicability.

## 6   Conclusion

We regard our work as a small step towards improving the credit assignment problem when dealing with augmented action space by better using the geometry or imposing structure on enlarged action space. Given a similarity metric, MASP allows similar actions to *move together*, effectively one learning from the other. We further demonstrate that the similarity metric can be learned efficiently and simultaneously with the policy, exploiting the meta-gradient framework. The overall proposed method MASP leads to effective and robust use of macro-actions, and improvements on Atari and StreetFighter. We believe our framework can be expanded further either for improving interpretability of action spaces or to exploit more state or policy dependent similarities.

**Acknowledgements.**   This research was supported by the project "Romanian Hub for Artificial Intelligence - HRIA", Smart Growth, Digitization and Financial Instruments Program, 2021-2027, MySMIS no. 351416.

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

# A MASP Regularization and Implementation Details

## A.1 A.1 Detailed Formulation of MASP

The Macro-Action Similarity Penalty (MASP) is a regularization term designed to enforce smoothness across the Q-values of similar actions (both primitive and macro-actions) in the augmented action space. To achieve this, we define a similarity matrix $\Sigma \in \mathbb{R}^{|\mathcal{A}| \times |\mathcal{A}|}$, where each entry $\Sigma_{ij} \geq 0$ represents the degree of similarity between actions $a_i$ and $a_j$. In our implementation, $\Sigma$ is constrained to be symmetric and non-negative, but we do not require it to be positive-definite.

Given a batch of transitions $\{(s_i, a_i, r_i, s_{i+1})\}_{i=1}^{n}$, MASP adds the following regularization to the TD loss:

$$\mathcal{L}_{\text{MASP}} = \eta \cdot \frac{1}{n} \sum_{i=1}^{n} \|Q(s_i, \cdot; \theta) - \Sigma Q(s_i, \cdot; \theta)\|_2^2 \tag{3}$$

where $Q(s_i, \cdot; \theta) \in \mathbb{R}^{|\mathcal{A}|}$ is the vector of Q-values for all actions at state $s_i$, and $\eta$ is a tunable regularization coefficient.

**Intuition:** This penalty encourages the Q-values of similar actions (according to $\Sigma$) to be close, effectively sharing credit among macro-actions that are functionally related. Unlike options or hierarchical RL, our penalty is "soft"—allowing some dispersion for exploration and differentiation.

—

## A.2 A.2 Meta-learning Procedure for $\Sigma$

The similarity matrix $\Sigma$ is meta-learned jointly with the agent's main network parameters $\theta$ via meta-gradients. This is achieved as follows:

1. **Inner Update (Agent Step):**

- Sample a trajectory $\tau$ from the replay buffer.

- Perform a standard TD update with the MASP penalty, updating $\theta \rightarrow \theta'$ using $\Sigma$ fixed.

2. **Outer Update (Meta Step):**

- Sample a new trajectory $\tau'$.

- Evaluate the performance of the updated $\theta'$ using a meta-objective (the standard TD loss).

- Compute the meta-gradient of this meta-objective w.r.t. $\Sigma$ (backpropagating through the inner update step).

- Update $\Sigma$ with a separate learning rate $\beta$.

This approach is similar to the meta-gradient method introduced by Xu et al. [32]. In practice, we use automatic differentiation and checkpointing to efficiently compute $\frac{d\theta'}{d\Sigma}$.

**Algorithmic Details:**

- See Algorithm 1 in the main text for the pseudocode.

- $\Sigma$ is parameterized as a free matrix with symmetry enforced by averaging with its transpose after each update.

- To prevent divergence, we clip the entries of $\Sigma$ to the range $[0, 1]$ after each update.

—

## A.3 A.3 Practical Implementation Details

**Embedding $\Sigma$:** For large action spaces, representing $\Sigma$ explicitly can be expensive. We instead flatten $\Sigma$ and use a learned projection:

$$e_\Sigma = W_{\text{emb}} \cdot \text{vec}(\Sigma)$$

where $W_{\text{emb}}$ is a trainable matrix, and $e_\Sigma$ is a low-dimensional embedding vector. This is concatenated with the state embedding and fed into the Q-network.

**Additional Training Details:**

- Both $\Sigma$ and $W_{\mathrm{emb}}$ are updated via backpropagation from the main loss.

- The meta-objective for $\Sigma$ updates does not include gradients through $W_{\mathrm{emb}}$.

- To prevent $\Sigma$ from degenerating to the identity or to a rank-one matrix, we add a small entropy penalty to its row-normalized version during meta-learning.

**Computational Cost:**

- The MASP regularization is vectorized using batched matrix multiplications.

- The additional overhead is roughly $20\%$ over standard DQN (wallclock time), dominated by meta-gradient computation and matrix operations.

# B    Experimental Details and Hyperparameters

We summarize Atari preprocessing settings in Table 6 and the main algorithm hyperparameters in Table 7.

| Hyperparameter | Value |
| --- | --- |
| Max frames per episode | 108,000 |
| Observation down-sampling | (84, 84) |
| Num. action repeats | 4 |
| Num. stacked frames | 1 |
| Terminal state on loss of life | *true* |
| Random noops range | 30 |
| Sticky actions | *true* |
| Frames max pooled | 3 and 4 |
| Grayscaled/RGB | Grayscaled |
| Action set | Full |

Table 6: Atari pre-processing hyperparameters.

**Reproducibility and Code Availability**

To facilitate reproducibility, all code used for the experiments and MASP implementation is available at: `https://github.com/rl-submissions/macro-credit-masp`.

## B.1    Compute Resources and Training Time

Our experiments were conducted on a compute cluster equipped with 192GB RAM and a combination of NVIDIA GPUs: 1x RTX 5090, 2x RTX 4090, and 4x RTX 3090 cards. All Atari experiments typically required approximately one week to complete per full experimental run (including all seeds, ablations, and sweeps). Training runs for StreetFighter environments generally completed in approximately two days.

# C    Additional Experimental Results

## C.1    Atari Results

Full Atari performance for all methods and human-normalized scores are reported in Table 8.

## C.2    Streetfighter II Experimental Details

Streetfighter II experiments were conducted with domain-specific preprocessing and macro-action settings, including a reduced action set and macro-actions corresponding to common combos. All environment and training hyperparameters are detailed in Table 9.

| Hyperparameter | Value |
|---|---|
| Optimizer | Adam |
| Learning rate | $6.25 \times 10^{-5}$ |
| Batch size | 32 |
| Discount factor $\gamma$ | 0.99 |
| Replay buffer size | $1 \times 10^{6}$ |
| Target network update period | 10,000 steps |
| Gradient clipping | 10 |
| Exploration $\epsilon$ (initial / final) | 1.0 / 0.01 |
| Exploration decay schedule | 1M steps |
| Multi-step returns ($n$) | 3 |
| Noisy nets | *true* |
| Distributional atoms | 51 |
| Distributional min/max values | -10 / 10 |
| Dueling network | *true* |
| Prioritized replay $\alpha$ | 0.5 |
| Prioritized replay $\beta$ | $0.4 \rightarrow 1.0$ |
| Macro-action set size ($k$) | 32 (see ablations) |
| Macro-action length | 3–8 |
| MASP penalty weight $\eta$ | 0.1, 0.3, 0.5, 0.7, 1 (swept) |
| Meta-learning rate $\beta$ | 0.001, 0.005, 0.01 (swept) |
| $\Sigma$ embedding dim ($e_\Sigma$) | 32 |

Table 7: Main hyperparameters used for Rainbow-DQN, macro-action augmentation, and MASP regularization in all experiments.

A visualization of the learned similarity matrix $\Sigma$ for macro-actions in Street Fighter II can be found in Figure 3 in this appendix, which illustrates the clustering of related macro-actions discovered by the agent during training.

## C.3 MiniGrid Experiments

**Experimental Setup.** We also tested MASP on MiniGrid [37] environments designed to require planning and structured exploration. We selected tasks such as DoorKey, FourRooms, LockedRoom, and ObstructedMaze-Full.

**Implementation Details.** Macro-actions consist of fixed sequences derived from typical interaction patterns (e.g., forward-forward-turn). We compare RAINBOW DQN, RAINBOW with macro-actions, and RAINBOW with MASP. Each agent is trained for 500,000 steps. Hyperparameter tuning for $\eta$ was done per environment.

**Results and Analysis.** As shown in Table 10, MASP improves success rates over both baselines across all tasks. In simpler settings, the improvements are significant, and in more complex environments like ObstructedMaze-Full, MASP is critical to achieving high success.

## C.4 Transferability

Another key motivation for the Macro-Action Similarity Penalty (MASP) framework is to improve transferability across tasks and environments. In principle, MASP regularization encourages the agent to learn more robust and generalizable representations by smoothing the Q-values among similar macro-actions, potentially facilitating adaptation to new tasks where action semantics overlap.

**Transfer Protocol:** In our experiments, we evaluated transferability by taking agents trained with MASP on a subset of tasks and fine-tuning them (with or without additional MASP updates) on related environments. In practice, the learned similarity matrix $\Sigma$ and macro-action set can be reused or adapted for downstream tasks, reducing the need for retraining from scratch.

| Game | Rainbow-DQN | +Macro-Actions | +MASP Score | HN Score |
|---|---|---|---|---|
| Alien | 6,022.9 ± 718.2 | 3,714.8 ± 246.0 | **7614.4 ± 388.5** | (121.4) |
| Amidar | 202.8 ± 23.4 | 185.8 ± 8.0 | **272.5 ± 7.0** | (48.7) |
| Assault | 14,491.7 ± 759.0 | 11,368.2 ± 1174.3 | **15665.7 ± 1682.2** | (540.4) |
| Asterix | 280,114.0 ± 23760.7 | 225,501.1 ± 25686.6 | **356834.2 ± 22386.9** | (2102.2) |
| Asteroids | 2,249.4 ± 191.5 | 2,093.2 ± 148.5 | **3544.9 ± 362.3** | (52.3) |
| Atlantis | 814,684.0 ± 42022.1 | 766,181.5 ± 52739.2 | **933826.3 ± 48727.3** | (6721.3) |
| Bank Heist | 826.0 ± 60.3 | 708.5 ± 20.8 | **1159.4 ± 70.6** | (219.2) |
| Battle Zone | 52,040.0 ± 4502.2 | 30,290.9 ± 2185.3 | **67537.1 ± 5791.1** | (398.2) |
| Beam Rider | 21,768.5 ± 1356.5 | 16,221.3 ± 1208.4 | **29374.2 ± 3309.8** | (232.5) |
| Berzerk | 1,793.4 ± 96.1 | 1,026.2 ± 36.7 | **2635.8 ± 122.0** | (180.3) |
| Bowling | 39.4 ± 4.2 | 35.1 ± 2.6 | **66.6 ± 6.0** | (34.2) |
| Boxing | 54.9 ± 2.7 | 50.6 ± 1.5 | **100.0 ± 10.3** | (1000.0) |
| Breakout | 379.5 ± 25.1 | 252.9 ± 22.2 | **884.4 ± 74.0** | (1011.2) |
| Centipede | 7,160.9 ± 211.7 | 4,487.6 ± 125.5 | **7489.6 ± 445.0** | (72.4) |
| Chopper Command | 10,916.0 ± 1034.1 | 7,464.7 ± 392.6 | **11592.2 ± 876.8** | (73.3) |
| Crazy Climber | 143,962.0 ± 13254.7 | 80,474.4 ± 7541.7 | **158672.8 ± 7774.1** | (207.5) |
| Defender | 47,671.3 ± 4264.2 | 30,844.1 ± 943.5 | **58679.4 ± 7205.4** | (346.3) |
| Demon Attack | 109,670.7 ± 2672.4 | 72,716.0 ± 5913.2 | **117663.3 ± 11120.4** | (1014.3) |
| Double Dunk | -0.6 ± 0.0 | -0.7 ± 0.1 | **-0.2 ± 0.0** | (53.7) |
| Enduro | 2,061.1 ± 83.3 | 1,324.7 ± 148.7 | **2266.6 ± 58.3** | (67.6) |
| Fishing Derby | 22.6 ± 2.8 | 22.6 ± 0.6 | **36.9.6 ± 1.0** | (75.3) |
| Freeway | 29.1 ± 2.7 | 25.2 ± 1.3 | **30.3 ± 3.7** | (101.0) |
| Frostbite | 4,141.1 ± 175.4 | 3,361.2 ± 356.4 | **5566.7 ± 317.3** | (65.5) |
| Gopher | 72,595.7 ± 6706.9 | 45,801.3 ± 5387.9 | **78992.5 ± 3703.2** | (803.5) |
| Gravitar | 567.5 ± 20.5 | 367.6 ± 42.6 | **645.5 ± 31.8** | (60.5) |
| Hero | 50,496.8 ± 3850.7 | 42,269.0 ± 1228.2 | **62730.1 ± 3469.7** | (261.0) |
| Ice Hockey | -0.7 ± 0.1 | -0.8 ± 0.0 | **-0.1 ± 0.0** | (53.6) |
| Kangaroo | 10,841.0 ± 1247.0 | 8,271.9 ± 1008.4 | **11225.7 ± 1304.9** | (132.6) |
| Krull | 6,715.5 ± 823.5 | 4,597.9 ± 342.5 | **8251.1 ± 858.9** | (65.1) |
| Kung Fu Master | 28,999.8 ± 2080.4 | 18,165.9 ± 1865.3 | **36837.2 ± 1430.2** | (151.2) |
| Montezuma's Revenge | 154.0 ± 16.9 | 124.0 ± 5.3 | **400.0 ± 18.1** | (3.5) |
| Ms Pacman | 2,570.2 ± 204.6 | 1,471.5 ± 185.3 | **2966.5 ± 360.5** | (67.6) |
| Name This Game | 11,686.5 ± 315.2 | 7,252.3 ± 180.3 | **12745.8 ± 685.4** | (131.5) |
| Phoenix | 103,061.6 ± 10294.9 | 89,925.5 ± 10494.5 | **116444.7 ± 5868.9** | (231.2) |
| Pitfall | -37.6 ± 2.0 | -49.8 ± 4.8 | **-12.8 ± 1.1** | (3.3) |
| Pong | **19.0 ± 1.4** | **19.0 ± 1.0** | **19.0 ± 0.3** | (104.2) |
| Private Eye | 1,704.4 ± 41.7 | 1,358.2 ± 94.9 | **2244.3 ± 150.9** | (13.2) |
| Q Bert | 18,397.6 ± 637.1 | 12,276.2 ± 854.1 | **22774.8 ± 1373.3** | (417.0) |
| Road Runner | 54,261.0 ± 2330.3 | 45,952.7 ± 1687.5 | **62633.7 ± 3796.5** | (220.4) |
| Robotank | 55.2 ± 6.4 | 50.5 ± 6.3 | **64.5 ± 7.6** | (75.1) |
| Seaquest | 19,176.0 ± 1428.9 | 13,917.1 ± 641.1 | **23768.8 ± 1476.9** | (240.0) |
| Skiing | -11,685.8 ± 787.3 | -15,069.0 ± 566.2 | **-10114.6 ± 575.5** | (42.6) |
| Solaris | 2,860.7 ± 153.5 | 1,764.4 ± 217.6 | **4488.9 ± 267.5** | (75.3) |
| Space Invaders | 12,629.0 ± 1413.2 | 7,496.4 ± 532.9 | **16668.2 ± 2050.5** | (197.6) |
| Star Gunner | 123,853.0 ± 4413.1 | 112,043.0 ± 12667.8 | **169778.8 ± 11049.3** | (533.5) |
| Surround | 7.0 ± 0.3 | 6.3 ± 0.3 | **8.72 ± 0.72** | (69.8) |
| Tennis | -2.2 ± 0.2 | -3.0 ± 0.4 | **12.6 ± 0.6** | (60.6) |
| Time Pilot | 11,190.5 ± 410.0 | 8,331.4 ± 931.3 | **15583.3 ± 707.4** | (209.7) |
| Tutankham | 126.9 ± 5.0 | 70.5 ± 7.9 | **179.6 ± 14.9** | (48.2) |
| Venture | 45.0 ± 3.4 | 25.0 ± 3.1 | **133.7 ± 8.4** | (20.5) |
| Video Pinball | 506,817.2 ± 16888.8 | 290,039.4 ± 11557.8 | **577339.3 ± 20750.9** | (327.4) |
| Wizard of Wor | 14,631.5 ± 948.6 | 10,125.1 ± 891.3 | **18866.5 ± 1771.3** | (259.5) |
| Yarr's Revenge | 93,007.9 ± 5494.8 | 55,584.7 ± 3232.2 | **103544.9 ± 3201.0** | (134.8) |
| Zaxxon | 19,658.0 ± 2229.6 | 17,553.0 ± 1147.8 | **26566.7 ± 2208.2** | (103.3) |

Table 8: Comparison between RAINBOW DQN [11], RAINBOW-DQN with macro-actions (+Macro-Actions), and RAINBOW DQN with macro-actions similarity penalty (+MASP). **Bold** indicates maximal raw performance between RAINBOW DQN and MASP. Human-normalized (HN) scores are shown in parentheses. Cells highlighted in pink denote HN ≥ 100.

| Hyperparameter | Value |
|---|---|
| Observation shape | (128, 128, 3) |
| Frame skip / Action repeat | 2 |
| Num. stacked frames | 4 |
| Reward clipping | [-1, 1] |
| Opponent | Random / AI Level 3 |
| Max episode steps | 18,000 |
| Terminal on round loss | *true* |
| No-op start range | 0–10 |
| Sticky actions | *false* |
| Action set | Reduced (15 discrete moves) |
| Combo macro-actions | *true* |
| Macro-action set size ($k$) | 24 |
| Macro-action length | 2–6 |
| MASP penalty weight $\eta$ | 0.3, 0.5 (swept) |
| Meta-learning rate $\beta$ | 0.001, 0.003 (swept) |
| $\Sigma$ embedding dim ($e_\Sigma$) | 16 |
| Optimizer | Adam |
| Learning rate | $1 \times 10^{-4}$ |
| Batch size | 32 |
| Replay buffer size | $5 \times 10^5$ |
| Discount factor $\gamma$ | 0.99 |
| Target network update period | 5,000 steps |
| Exploration $\epsilon$ (initial/final) | 1.0 / 0.05 |
| Exploration decay schedule | 200k steps |
| Dueling network | *true* |
| Distributional RL | *true* |

Table 9: Hyperparameters for Streetfighter II experiments. Settings reflect domain-specific differences, including observation size, combo macro-actions, and action set.

| Task | Rainbow DQN | Macro-Actions | MASP |
|---|---|---|---|
| **Door Key** | 0.83 | **0.88** | 1 |
| **Four Rooms** | 0.69 | **0.87** | 1 |
| **Locked Room** | 0.58 | **0.77** | 0.97 |
| **Obstructed Maze Full** | 0.57 | 0.84 | **0.91** |

Table 10: Comparison between the success rate of RAINBOW DQN, RAINBOW DQN with macro-actions and MASP for MiniGrid environments.

**Observations:** We found that MASP-trained agents generally adapted more quickly and achieved higher initial performance on transfer tasks compared to standard Rainbow DQN baselines. This suggests that MASP helps encode transferable structure in the Q-function and macro-action embeddings.

**Limitations:** The degree of transfer benefit depends on the similarity between source and target task action spaces. Large discrepancies may require re-learning or adaptation of $\Sigma$.

| Hyperparameter | Value |
|---|---|
| Max frames per episode | 2,000 |
| Num. action repeats | 1 |
| Num. stacked frames | 1 |
| Terminal state on loss of life | N/A |
| Random noops range | 0 |
| Sticky actions | *false* |
| Replay buffer size | $5 \times 10^4$ |
| Batch size | 64 |
| Learning rate | $1 \times 10^{-4}$ |
| Discount factor $\gamma$ | 0.99 |
| Target network update period | 1,000 steps |
| Exploration $\epsilon$ (initial / final) | 0.2 / 0.01 |
| Exploration decay schedule | 50k steps |
| Multi-step returns ($n$) | 1 |
| Noisy nets | *false* |
| Dueling network | *false* |
| Macro-action set size ($k$) | 8 |
| Macro-action length | 2–4 |
| MASP penalty weight $\eta$ | 0.05, 0.1, 0.3 (swept) |
| Meta-learning rate $\beta$ | 0.001 |
| $\Sigma$ embedding dim ($e_\Sigma$) | 8 |

Table 11: MiniGrid-specific hyperparameters. Only hyperparameters that differ from Atari are shown.

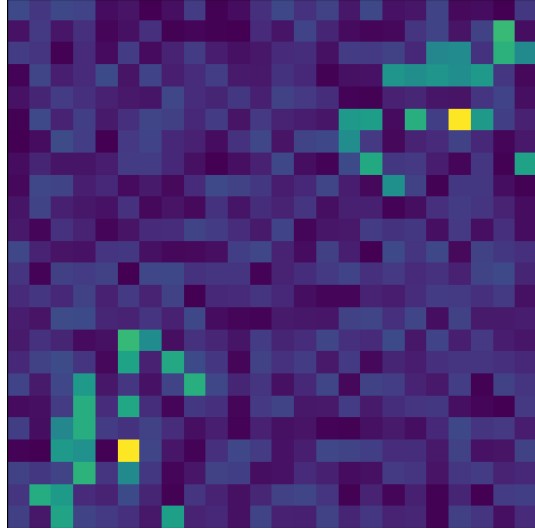

Figure 3: Sample $\Sigma$ matrix from Street Fighter II experiments, illustrating the learned similarities between different macro-actions. Distinct clusters with higher values indicate groups of macro-actions that are functionally related or often co-activated. In contrast, the regions of the matrix with the lowest values and lacking visible structure correspond to primitive actions, which are entirely independent and dissimilar to each other.

