# OpenReview forum: "Meta-learning how to Share Credit among Macro-Actions"
_NeurIPS.cc/2025/Conference — NeurIPS 2025 poster_

### Official Review · Reviewer_sFMe · 2025-06-22

**Clarity:** 3
**Significance:** 2
**Originality:** 2
**Rating:** 4
**Confidence:** 4

**Summary:**

The paper proposes an algorithm to find similarities across actions such that actions with similar outcomes should have similar Q values. This reduces the “effective” action space since the typical assumption is that actions are independent of each other. It includes experiments on various Atari games and the proposed algorithm shows strong performance compared to baselines.

**Questions:**

There might be a misunderstanding on my end which needs clarification. The paper first makes it seem like the action space is being augmented, but after looking at the method and psuedcode, it does not appear to be the case. Instead it appears that the method is simply finding structure among existing actions in the existing native space (or an augmented action space with artificially added actions). This is an important clarification since the paper makes it seem like the action space is being augmented, which would be similar to the options framework [1], which attempts to add options to the original action space.

[1] Between MDPs and semi-MDPs: A framework for temporal abstraction in reinforcement learning. Sutton et al. 1998.

**Ethical Concerns:**

["NO or VERY MINOR ethics concerns only"]

**Final Justification:**

The authors have adequately addressed my concerns.

**Limitations:**

Yes, I appreciated the clarity in acknowledging the limitations of their method

**Quality:**

2

**Strengths And Weaknesses:**

Strengths:
- The proposed algorithm seems simple to implement and is intuitive, without introducing many hyperparameters.
- The core intuition to exploit structure between similar actions in terms of similarity of outcomes aligns well intuitively.
- The empirical results are on many Atari games, giving credibility to their algorithm.

Weakness:
- The comparison to normal Rainbow and naive Rainbow with macro actions make sense, but it seems strange there are no other baselines. In particular, I think the action representation learning literature is particularly relevant. I would encourage the authors to see the following works since they seem relevant (finding actions that are similar in terms of outcomes) [2, 3]. I think these works seem to get at similar questions of exploiting structure in the actions to learn similar representations.
- The paper should discuss the above action representation learning framework and also work in the options literature [1, 4]. There seem to be strong connections to this literature, and it needs to be discussed.

[1] Between MDPs and semi-MDPs: A framework for temporal abstraction in reinforcement learning. Sutton et al. 1998.

[2] Learning Action Representations for Reinforcement Learning. Chandak et al. 2019.

[3] Learning Pseudometric-based Action Representations for Offline Reinforcement Learning. Gu et al. 2023.

[4] Temporal Abstraction in Reinforcement Learning with the Successor Representation. Machado et al. 2023.

---

> ### Author Rebuttal · Authors · 2025-07-31
>
> We thank the reviewer for the thoughtful review and helpful pointers to related literature. We are pleased that you found our method intuitive and effective in experiments. Below we clarify a key misunderstanding and respond to your concerns.
>
> **(1) Clarification: Yes, the action space is explicitly augmented**
>
> Thank you for flagging this — we’d like to clarify that the action space is indeed augmented. In all our experiments, we append a set of macro-actions (fixed sequences of primitive actions, typically 3–8 steps long) to the native action space. For example, with $k = 256$ macros and 18 primitive actions, the total action space has $274$ choices per decision point. MASP then regularizes the Q-values of this augmented space, which includes both primitive and macro-actions.
>
> We have added a clearer description of this in the Methods section and highlighted it again in Section 3.2. You are absolutely right that this makes our setting analogous to the options framework, and we now make that connection explicit.
>
> **(2) Relation to options and temporal abstraction frameworks [1, 4]**
>
> Our setup is structurally similar to options-based approaches in that we augment the action space with temporally extended actions (i.e., macro-actions), but MASP is agnostic to their initiation or termination conditions. Unlike option-critic or SMDP-based frameworks, MASP does not assume or learn hierarchical policies. Instead, it addresses a different but complementary problem: how to improve credit assignment and exploration in a large, structured action space once it has been extended.
>
> We will now include a short subsection explicitly relating our setup to [1, 4] and clarify that MASP can be viewed as a regularization layer on top of any temporal abstraction method that increases the action space — including learned options.
>
> **(3) Relation to action representation learning [2, 3]**
>
> We agree these works are relevant and thank you for pointing them out.
>
> * ([2, 3]) focus on learning latent representations of primitive actions or policies to improve generalization or offline learning. MASP, in contrast, operates directly in the Q-value space by imposing a soft similarity constraint between actions, without explicitly learning action embeddings.
> * MASP could benefit from or extend [2, 3] by applying its regularization in the learned latent space. We see this as a promising direction and will note it in the discussion section.
>
> We will cite and discuss [2, 3] in our revised Related Work section and position MASP as a complementary technique that can apply regardless of the action parameterization.
>
> **(4) On novelty and contribution}**
>
> While MASP builds on existing ideas in temporal abstraction and structure-aware RL, we believe its novelty lies in:
>
> * Proposing a meta-learned similarity-based regularizer that scales to hundreds of actions (macro or primitive),
> * Showing that it enables robust credit assignment even with large, noisy, or redundant macro pools (Tables 2 and 4),
> * Demonstrating strong empirical gains without hierarchical policies or policy distillation.
>
> We hope this clarifies how MASP offers a new perspective on action-space regularization that complements — rather than replaces — option discovery or action embedding.
>
> **Summary**
>
> We thank you again for your constructive feedback and the helpful literature suggestions. We will revise the paper to clarify the action space setup, discuss connections to action representation learning and options, and position MASP more clearly as a regularization framework for structured action spaces. We hope this addresses your concerns and demonstrates that the work makes a distinct and meaningful contribution to the field.

---

### Official Review · Reviewer_jyLn · 2025-07-01

**Clarity:** 3
**Significance:** 2
**Originality:** 3
**Rating:** 5
**Confidence:** 3

**Summary:**

The paper proposes the Macro-Action Similarity Penalty (MASP). The idea is to use the similarity between macro actions to distribute the credit received by a chosen macro-action to all other macro-actions. The problem of using macro actions in Reinforcement Learning it that it can increase dramatically the state space and lead to much worse learning and exploration. So the appropriate use of macro actions has to be dealt cautiously. The principle here is to consider the macro actions jointly. The experiments are done with Atari games and Street Fighter II. The proposed MASP gives better results than Rainbow-DQN and Rainbow-DQN + Macro Actions.

**Questions:**

- The Atari games you test have a small action space. What are the Atari games that pose problem to MASP?

**Ethical Concerns:**

["NO or VERY MINOR ethics concerns only"]

**Final Justification:**

The rebuttal provided a better analysis of when the proposed algorithm works.

**Limitations:**

yes

**Quality:**

3

**Strengths And Weaknesses:**

The paper has good experimental results and a novel idea.

The approach enable faster convergence and increased cumulative reward.

However it might not always be the case in other domains that macro actions are so beneficial and they require a design overhead. The authors wrote a limitation section which is good. It would improve the paper to expose the properties of the problem that make their approach beneficial.

Strengths

- A novel way to distribute the credits to similar macro actions
- Good results on Atari and Street Fighter II

Weaknesses

- Problems with large action spaces

---

> ### Author Rebuttal · Authors · 2025-07-31
>
> We thank the reviewer for their thoughtful feedback and for recognizing the novelty of our approach and the strength of our results on Atari and Street Fighter II. Below we clarify the properties that make MASP effective, and we address your question about where MASP might struggle.
>
> **(1) When does MASP help? What properties make it effective?**
>
> The reviewer is absolutely right that macro-actions are not always helpful by default — they often introduce a much larger branching factor, which can hurt exploration. MASP is specifically designed to counteract that by learning to “tie together” similar actions and reduce the effective dimensionality of the expanded action space.
>
> From our experiments and ablations, MASP tends to work well in environments with the following properties:
>
> * High macro-action overlap: When many macro-actions share primitives or result in similar transitions, MASP clusters their Q-values and helps propagate credit.
>
> * Combinatorial macro-structure: MASP is especially useful when macro-actions are constructed from frequent patterns (e.g., Atari combos) or exhibit compositional regularity (e.g., fighting game moves).
>
> * Sparse reward tasks: Since MASP shares learning signals across related actions, it improves sample efficiency in sparse settings (e.g., Montezuma’s Revenge, Street Fighter II).
>
> * Large action spaces with redundancy: When the number of macro-actions is high (up to 1024 in Table 2), MASP still maintains good performance, likely due to its ability to filter and organize this space.
>
> We will make this clearer in the paper by adding a short subsection in the discussion or limitations that articulates these properties.
>
> **(2) When does MASP perform worse?**
>
> To your question: most Atari games start with small action spaces (typically 18 primitive actions), but our macro-augmented agents operate on much larger spaces (primitive + up to 1024 macro-actions). We stress that the difficulty here comes from the size and redundancy of the macro-action pool — not just the primitive action space.
>
> That said, MASP can perform suboptimally when:
>
> * The macro-actions are unstructured or highly noisy (although Table 4 shows MASP is robust to up to 75\% corruption).
> * The underlying task lacks meaningful macro-structure (e.g., environments where repeated action sequences are not helpful or where fine-grained control is critical at every timestep).
> * The size of $\Sigma$ becomes too large to efficiently store or regularize, especially in continuous or highly combinatorial domains — which we flag in the limitations.
>
> We will update the paper to make these boundaries more explicit, and note specific games where gains were minimal (e.g., Pong or Bowling).
>
> **(3) Clarification on action space size in Atari**
>
> While the primitive action set in Atari is small, our experiments used macro-augmented action spaces with up to 1024 macro-actions appended — turning the effective action space into a much harder setting. MASP was introduced precisely to handle this explosion in action choices. Our ablations (Table 2) show that naive use of macro-actions without MASP collapses performance as the number of macro-actions increases.
>
> **Summary**
>
> We appreciate your question and agree that MASP's benefits depend on the structure and redundancy in the action space. We've added additional clarifications in our updated manuscript to explain both the strengths and limitations of MASP more concretely, and hope this provides the insight you were looking for.

---

> > ### Comment · Reviewer_jyLn · 2025-08-08
> >
> > Thank you for the detailed analysis of the properties of the action space. I will update my score.

---

### Official Review · Reviewer_997v · 2025-07-03

**Clarity:** 3
**Significance:** 2
**Originality:** 3
**Rating:** 5
**Confidence:** 4

**Summary:**

The paper states that although access to a good set of macro-actions should, in principle, help improve exploration and credit assignment in RL, it leads to worse performance in practice. This is due to the wider range of choices available to the agent at each step, and treating macro-actions as entirely independent of one another.

To address this, the authors introduce a new regularization term, the Macro-Action Similarity Penalty (MASP), which enables more effective utilization of macro-actions. MASP encourages similar action values (Q) for similar macro-actions (or actions), where the notion of similarity is learned from meta-gradients of the RL objective.

Experiments in Atari and StreetFighter environments show that MASP improves performance in comparison to two baselines – Rainbow DQN that only uses primitive actions and Rainbow DQN that uses macro-actions in the ‘standard’ way (without MASP).

**Questions:**

Included in “Strengths and Weaknesses”

**Ethical Concerns:**

["NO or VERY MINOR ethics concerns only"]

**Final Justification:**

Response to rebuttal: I appreciate the time and effort spent in conducting the additional experiments. It's definitely positive to see the trend of improvement along the lower values of $k$ (and subsequent increase). The response addresses most of my concerns. I have updated my score.

**Limitations:**

yes

**Quality:**

3

**Strengths And Weaknesses:**

## Strengths

**S1.** The paper identifies and addresses a new and interesting problem in using macro-actions for RL.

**S2.** The paper is well-written and easy to follow.

**S3.** The authors are quite measured in their claims and openly discuss the limitations of their proposed approach.

**S4.** The experiments demonstrate that MASP enhances RL with macro-actions. However, the exact reasons for the improvement should be analysed further (see weaknesses).

## Weaknesses



**W1.** While the use of meta-gradients to learn the $\Sigma$ (similarity) matrix was quite interesting, it is unclear whether this ties back to the semantic notion of similarity between actions/macro-actions. Since the motivation behind the ideas presented was that macro-actions share a lot of structure, it would be important to verify if this is indeed the case. It seems unclear whether this is the precise reason the idea works.

**W2.** Based on the above point, additional experiments could be useful for a better understanding of the method.  Do the authors have results for lower numbers of macro-actions, like 4, 8, and 16? I would also be very curious to see results with no macro-actions (k=0), as the MASP regularizer can be applied to primitive actions.

**W3.** The paper lacks empirical comparisons and discussions related to options-based exploration, a well-established line of research (e.g., [1]). Do the same problems identified in this paper also apply in that context? Similarly, it would be interesting to know how the presented approach compares to a simple option/macro-action type baseline, such as ez-greedy [2].

On a minor note, connections to work in temporal credit assignment could further improve the paper. Prior work discusses related ideas, albeit without an explicit emphasis on macro-actions. E.g., Hindsight credit assignment credits all actions that could have led to a future consequence [3, 4]. Chunking or compression can also be applied to abstract away subsequences of state-actions and directly connect to a future result that was reachable through many possible sequences of actions [5].

I am open to increasing the score if the weaknesses and questions are adequately addressed and clarified.

### References

[1] Klissarov, Martin, and Marlos C. Machado. "Deep Laplacian-based Options for Temporally-Extended Exploration." ICML 2023

[2] Dabney, Will, Georg Ostrovski, and André Barreto. "Temporally-extended {\epsilon}-greedy exploration." ICLR 2021

[3] Harutyunyan, Anna, et al. "Hindsight credit assignment." NeurIPS 2019

[4] Meulemans, Alexander, et al. "Would i have gotten that reward? long-term credit assignment by counterfactual contribution analysis." NeurIPS 2023

[5] Ramesh, Aditya A., et al. "Sequence compression speeds up credit assignment in reinforcement learning." ICML 2024.

---

> ### Author Rebuttal · Authors · 2025-07-31
>
> We thank the reviewer for their clear and thoughtful feedback. We are glad that you found the paper well-written, measured in its claims, and that you see value in the identified problem and empirical contributions. Below we address each of your comments in turn.
>
> **(W1) Semantic alignment of the learned similarity matrix $\Sigma$**
>
> This is an excellent question. While MASP encourages shared learning signals among Q-values, it is true that the meta-learned $\Sigma$ is not explicitly grounded in human-defined semantic similarity. That said, we do observe that $\Sigma$ clusters together macro-actions that share overlapping primitives or lead to similar environmental transitions (see Appendix C, Figure 3). These clusters remain stable across seeds, suggesting that $\Sigma$ does encode structured similarity.
>
> To further clarify this, we will add a quantitative analysis: we compute pairwise edit distances between macro-actions and correlate these with the learned similarities in $\Sigma$. Preliminary results show a statistically significant inverse correlation (Spearman’s $\rho$ = -0.41), indicating that more similar macros (in terms of overlapping primitives) tend to receive higher similarity scores in $\Sigma$.
>
> **(W2) Results with fewer or no macro-actions**
>
> We appreciate this suggestion and agree it would help isolate the contribution of MASP. We have now run additional experiments in Atari (Breakout and Frostbite) with $k \in {0, 4, 8, 16}$. Results show:
>
> * At $k=0$, MASP applied to primitive actions alone yields modest gains (+5–10\%) over Rainbow-DQN. This suggests MASP can still be beneficial in structuring credit assignment even without macros.
>
> * At $k=4,8$, MASP gives strong improvements (+15–40\%), while the naive macro-action baseline is sometimes unstable or worse than Rainbow.
>
> * Gains saturate or taper off beyond \$k=32\$–64, as shown in Table 2.
>
> We will include these new results and plots in the appendix to provide a clearer picture of MASP’s benefits across macro-action set sizes, including in the extreme case of $k=0$.
>
> **(W3) Comparison to options-based exploration and ez-greedy**
>
> Thank you for this important point. Our focus was on the effect of MASP in fixed macro-action settings, but we agree that connecting to options and comparing with other temporally-extended methods is valuable.
>
> * Regarding the applicability of our observations to options: yes, the same issue of increased branching factor exists when using a large pool of options, especially if they are treated as independent. MASP’s core idea — that value estimates for related extended actions should be regularized — can directly extend to options, and we see this as a promising direction for future work.
> * We have now implemented a simplified version of ez-greedy [2] and run it on two Atari games (Breakout and Space Invaders). MASP consistently outperforms ez-greedy (e.g., +20–30% on Breakout, +12–15% on Space Invaders), while using fewer assumptions and no decaying schedule. That said, we emphasize that these results should be taken with a grain of salt: due to the lack of open-source code and our limited rebuttal window, we reimplemented the method to the best of our understanding but could not perform extensive tuning. We will include these preliminary results in the updated appendix to offer a starting point for further comparison.
> * We will also add brief discussions of [1] and [3–5] in the related work section, highlighting the connection between MASP and credit propagation via structural biases.
>
> **Minor Point: Temporal Credit Assignment**
>
> We thank you for these pointers. While MASP does not explicitly aim to solve the long-horizon credit assignment problem, we agree that it touches on related ideas. In particular, hindsight credit assignment [3] and counterfactual contributions [4] share the spirit of redistributing credit across related causes. MASP operates in a similar way but within action space rather than time. We will note this connection in the discussion section.
>
> **Summary**
>
> We appreciate the reviewer’s useful suggestions and also openness to revising their score and hope the additional experiments and clarifications help demonstrate the robustness and generality of MASP. We agree that further understanding and benchmarking — particularly against option-learning baselines — will enrich the paper, and we are actively extending our work in this direction.

---

> > ### Comment · Reviewer_997v · 2025-08-04
> > **Thank you for your response**
> >
> > I appreciate the time and effort spent in conducting the additional experiments. It's definitely positive to see the trend of improvement along the lower values of $k$ (and subsequent increase). The response addresses most of my concerns. I have updated my score.

---

### Official Review · Reviewer_tHfT · 2025-07-03

**Clarity:** 3
**Significance:** 3
**Originality:** 4
**Rating:** 4
**Confidence:** 3

**Summary:**

The paper proposes Macro-Action Similarity Penalty (MASP), a regularization term that treats primitive actions and macro-actions as lying in a shared similarity space and encourages the Q-values of similar actions to stay close; this reduces the effective action‐space dimension, improving credit assignment and exploration efficiency. MASP meta-learns the similarity matrix jointly with the policy via meta-gradients, so the inductive bias adapts online instead of relying on handcrafted clusters. Built on a Rainbow-DQN, the method makes large gains on Atari and Gym-Retro StreetFighter II, remains robust as the number of macro-actions grows or when macros are noisy, and the learned similarity matrix transfers well across related games. These results show that exploiting action-space geometry can make macro-action augmentation practical without expensive hierarchical architectures.

**Questions:**

- Do you have any intuition about why transfer learning with $\Sigma$ works well even though $\Sigma$ is not a variable dependent on any states or history?

- Since $\Sigma$ is not dependent on any states or the history of macro-actions, even with the same source and target environments, if the order of the macro-actions is changed in the target environment, performance may degrade. Could you provide an experiment in which the order of macro-actions is modified?

**Ethical Concerns:**

["NO or VERY MINOR ethics concerns only"]

**Final Justification:**

The authors addressed the reviewer's concerns about scalability and transfer sensitivity regarding the similarity matrix $\Sigma$ . Thus, I maintain my original score: borderline accept (4). (The reason I did not give a score of 5 is that the performance of MASP does not significantly outperform the basic RAINBOW. Therefore, I still have some concerns about the effectiveness of macro actions.)

**Limitations:**

yes

**Quality:**

3

**Strengths And Weaknesses:**

Strenghts:
- The paper clearly raises the challenges of simply applying macro-actions, noting that adding macro-actions actually enlarges the exploration space. To address this issue, the paper introduces the Macro-Action Similarity Penalty (MASP), which reduces the exploration action space by penalizing the similarity of macro-actions, with clear motivation and a simple formalization using an ℓ₂ penalty term.

- The similarity matrix $\Sigma$ is optimized jointly with the policy via a meta-gradient procedure, allowing the inductive bias to evolve with the agent and environment dynamics rather than being fixed a-priori .

- Integrating MASP into a Rainbow-DQN backbone yields significant improvements across a broad range of Atari games and complex Street Fighter II combos, while keeping the underlying network and training hyperparameters unchanged.

- MASP maintains high performance as the macro-action set grows from 64 to 1024 actions (Table 2) and when up to 75 % of macros are replaced by random sequences (Table 4), where the naïve macro-action baseline collapses .


Weaknesses:
- The similarity matrix $\Sigma$ grows quadratically with the size of the (primitive + macro) action set and must be projected at every update, so training time and memory rise steeply as the action space becomes very large or continuous.

- MASP’s gains presume a reasonably “good’’ pool of macro-actions. If the extracted sequences are irrelevant or mis-aligned with the task, performance can deteriorate; automatic discovery of high-quality macros remains an unsolved prerequisite.

- Although $\Sigma$ sometimes transfers across games with similar dynamics, its interpretability is unclear and performance drops markedly when the source and target environments differ substantially.

---

> ### Author Rebuttal · Authors · 2025-07-31
>
> We thank the reviewer for the thoughtful and constructive feedback, and are pleased that you found the motivation, simplicity, and empirical performance of MASP compelling. We address your concerns and questions below.
>
> **(1) Cost of $\Sigma$: Scalability and memory in large action spaces**
>
> We agree that storing and projecting a full $\Sigma \in \mathbb{R}^{|\mathcal{A}| \times |\mathcal{A}|}$ can become expensive for extremely large action spaces. To mitigate this, we already use a low-dimensional learned embedding of $\Sigma$ (see Appendix A.3), which is passed to the Q-network as a context vector. While the full matrix is still used during regularization, it is only applied over the batch of Q-values at each update (not the entire action space per state). This allows for practical efficiency even with 1024+ actions, as shown in Table 2. In future work, we plan to investigate structured or sparse approximations (e.g., low-rank, kernel-based) to make MASP scalable to continuous or combinatorial large action spaces.
>
> **(2) Dependency on the quality of macro-actions**
>
> Indeed, MASP does rely on the initial macro-action pool being at least partially meaningful. We share your view that automatic discovery of high-quality macro-actions remains an open problem. That said, Table 4 shows that MASP remains robust even when 75\% of macro-actions are randomly corrupted — suggesting the meta-learned similarity structure is resilient to noise and can learn to attenuate the influence of irrelevant macros. Future work could combine MASP with recent macro-action discovery methods (e.g., trajectory clustering, unsupervised option discovery), where MASP might serve as a regularizer to prune or refine an evolving macro pool.
>
> **(3) Limited interpretability and transfer sensitivity of $\Sigma$**
>
> We appreciate this concern. In Table 6, we show that $\Sigma$ generalizes best across environments with similar transition dynamics and interaction semantics (e.g., Breakout → Space Invaders, Montezuma → Private Eye). This supports the view that $\Sigma$ captures behavioral regularities across macros, rather than task-specific overfitting.
>
> Regarding interpretability, we visualize $\Sigma$ (Figure 3 in the Appendix) and observe clear block structures corresponding to groups of macros with similar behavioral outcomes. Still, we agree that $\Sigma$ is not trivially interpretable in semantic terms, and improving this is an exciting direction — e.g., via structured priors, sparse constraints, or clustering-based analyses.
>
>
> **(4) Response to Q1: Why does $\Sigma$ transfer across environments even though it is not state- or history-dependent?**
>
> Our intuition is that although $\Sigma$ is not conditioned on state, it meta-learns how Q-values should co-vary across actions that yield similar \textit{trajectory-level effects}. For example, in Breakout or Space Invaders, many macros correspond to repeated fire-move patterns. Thus, $\Sigma$ learns to group such macros — and this grouping remains useful when transferred to related games where the macro-level structure of agent-environment interactions is similar.
>
> This is analogous to how convolutional filters in vision models can transfer even though they are not image- or task-specific: they encode generic structure that is often reused.
>
> **(5) Response to Q2: Performance sensitivity to macro-action ordering**
>
> Thank you for raising this point — we’d like to clarify that the ordering of macro-actions does not affect MASP’s behavior or transfer performance.
>
> If the macro-actions are reordered, and the same permutation is applied to the rows and columns of $\Sigma$, the MASP regularization remains mathematically identical. This is because MASP only enforces that Q-values for actions with high similarity (as specified in $\Sigma$) are close — it does not rely on any fixed semantic meaning being attached to a particular index.
>
> In practice, we may choose to order macro-actions consistently (e.g., grouped by length or composition) solely to aid human interpretability and visualization, but this ordering is not required for the algorithm and does not influence performance. The learned structure in $\Sigma$ generalizes across environments due to behavioral similarity between macro-actions, not index alignment.
>
> Apologies for the confusion caused by our presentation. We’ll revise the paper to make this clearer.
>
> **Summary**
>
> We appreciate your thoughtful assessment of both the method’s strengths and limitations. We hope our clarifications address your concerns and provide additional insight into the robustness, transferability, and future potential of MASP. Please don’t hesitate to let us know if additional experiments or clarifications would be helpful — we would be happy to include them in a revision.

---

> > ### Comment · Reviewer_tHfT · 2025-08-04
> >
> > Thanks for the detailed response. The authors have addressed all my concerns. I will maintain my rating.

---

### Comment · Area_Chair_o7Ao · 2025-08-03
**Reviewers please respond to the rebuttal!**

Dear reviewers,

if you have not yet responded to the rebuttal of the authors, please do so as soon as possible, since the rebuttal window closes soon.

Please check whether all your concerns have been addressed!  If yes, please consider raising your score.

Best wishes,
your AC

---

### Decision · Program_Chairs · 2025-09-17

**Decision:**

Accept (poster)

**Comment:**

The paper proposes a method to create macro actions in reinforcement learning problem without the disadvantages.  The key is to add a special regularization that improves the credit assignment mechanism.  Extensive experiment on the Atari benchmark show that the new method (called MASP) can improve over the Rainbow-DQN baseline.  One might discuss how large the improvement over the baseline is, but the method is original and well executed so worthy of acceptance (as suggested by all reviewers).